# Spike Gene Analysis and Prevalence of Porcine Epidemic Diarrhea Virus from Pigs in South Korea: 2013–2022

**DOI:** 10.3390/v15112165

**Published:** 2023-10-28

**Authors:** Gyu-Nam Park, Sok Song, SeEun Choe, Jihye Shin, Byung-Hyun An, Song-Yi Kim, Bang-Hun Hyun, Dong-Jun An

**Affiliations:** 1Virus Disease Division, Animal and Plant Quarantine Agency, Gimcheon 39660, Republic of Korea; changep0418@korea.kr (G.-N.P.); ssoboro@korea.kr (S.S.); ivvi59@korea.kr (S.C.); shinji227@korea.kr (J.S.); songkim@korea.kr (S.-Y.K.); hyunbh@korea.kr (B.-H.H.); 2College of Veterinary Medicine, Seoul University, Gwanak-ro, Gwanak-gu, Seoul 08826, Republic of Korea; anbh5043@gmail.com

**Keywords:** PEDV, evolution, piglet, diarrhea, spike, S INDEL

## Abstract

From late 2013–2022, 1131 cases of porcine epidemic diarrhea (PED) were reported to the Korean Animal Health Integrated System (KAHIS). There were four major outbreaks from winter to spring (2013–2014, 2017–2018, 2018–2019, and 2021–2022), with the main outbreaks occurring in Chungnam (CN), Jeonbuk (JB), and Jeju (JJ). Analysis of the complete spike (S) gene of 140/1131 KAHIS PEDV cases nationwide confirmed that 139 belonged to the G2b genotype and 1 to the G2a genotype. Among them, two strains (K17GG1 and K17GB3) were similar to an S INDEL isolated in the United States (strain OH851), and 12 strains had deletions (nucleotides (nt) 3–99) or insertions (12 nt) within the S gene. PEDVs in JJ formed a regionally independent cluster. The substitution rates (substitutions/site/year) were as follows: 1.5952 × 10^−3^ in CN, 1.8065 × 10^−3^ in JB, and 1.5113 × 10^−3^ in JJ. A Bayesian skyline plot showed that the effective population size of PEDs in JJ fell from 2013–2022, whereas in CN and JB it was maintained. Genotyping of 340 Korean PEDV strains, including the 140 PEDVs in this study and 200 Korean reference strains from GenBank, revealed that only the highly pathogenic non-INDEL type (G2b) was dominant from 2020 onwards. Therefore, it is predicted that the incidence of PED will be maintained by the G2b (non-INDEL) genotype.

## 1. Introduction

Porcine epidemic diarrhea virus (PEDV; genus, Alphacoronavirus; family, Coronaviridae; order, Nidovirales), an enteric pathogen with a genome of approximately 28 kb, has a negative impact on the swine industry [1,2]. The PEDV genome contains at least seven open reading frames (ORFs) encoding the ORF1a, ORF1b, S, ORF3, E, M, and N genes [3,4,5]. Two ORFs (OFR1a and ORF1b) encode replicase proteins required for viral replication and evasion of host immune responses [3,4]. The C terminus of the viral genome contains five ORFs encoding four structural proteins (spike protein (S), small envelope glycoprotein (E), membrane glycoprotein (M), and nucleocapsid protein (N)), as well as a hypothetical accessory protein, ORF3 [5]. The ectodomain of the S protein comprises S1 and S2 subunits, with S1 comprising N- and C-terminal domains (S1-NTD and S1-CTD) [6]. The role of the S1-CTD domain is to interact with aminopeptidase N (APN) to facilitate entry into target cells [7,8]. The S2 region is primarily responsible for triggering fusion of the viral envelope with the host cell membrane [9,10]. PEDV can infect pigs of all ages, but the highest mortality rate occurs in newborn pigs (around 1 week-of-age) and is due to acute diarrhea, vomiting, and dehydration [11,12]. PEDV first emerged in the late 1970s in the United Kingdom [13] and Belgium [1], followed by outbreaks in many European countries. In Asia, PEDV was first identified in China in 1984 and has since been detected in pig farms in Korea, Japan, and Thailand [14,15]. Although PED is endemic in most of these countries, a previous paper argued that the impact on the swine industry was not significant until 2010 [16], and that the pathogenicity of PED was weak in Korea, with no severe adverse effects on pig farms. However, a highly pathogenic strain of PEDV emerged in China in 2010, with a first outbreak in the United States in April 2013 that spread rapidly across the country [17]. Outbreaks in the United States, as well as other Asian countries, were reported immediately, but mortality rates in suckling piglets were as high as 100% [11,18,19]. This new, highly pathogenic mutant PEDV, which belongs to the G2b type, has caused severe diarrhea and high mortality in Korean piglets since the end of 2013, and outbreaks are particularly frequent in winter [20,21,22]. PEDV (type G2b) prevalent in the United States is categorized into two strains, as follows: the original PEDV (non-S INDEL) strain, which is highly pathogenic, and the S INDEL strain, which has reduced pathogenicity because of multiple deletions and insertions in the S1 subunit of the S protein [23]. Although not the same as the S INDEL strain prevalent in the United States, cases of INDEL strains with genetic mutations have also been reported in South Korea [24,25], including one strain with 15 nucleotides inserted in the S1 gene region (KOR/KNU-1601/2016) and two strains with 6 nucleotides deleted in the S2 gene region (KOR/KNU-1829/2018 and KOR/KNU- 1830/2018) [24,25]. In addition, genetic recombination between Korean PEDV genotypes was reported recently [26]. 

The purpose of the present study was to investigate the regional incidence, yearly genotype prevalence, and genetic variation of highly pathogenic G2b-type PEDV since its introduction into Korea in late 2013 up until 2022. In addition, we examined anti-PED antibody production in domestic sows in each of these years to investigate the relationship with PED outbreaks.

## 2. Materials and Methods

### 2.1. Detection of PEDV-Positive Samples from Suckling Piglets and Sows 

From November 2013 to December 2022, fecal and small intestine samples were collected from suckling pigs with diarrheal disease in 140 of the 1131 pig farms reported as PED-positive to the Korean Animal Health Integrated System (KAHIS). The KAHIS system (https://www.Kahis.go.kr, accessed on 3 April 2023) is operated by the central government, but livestock quarantine agencies in each local area carry out inspections, and important livestock infectious diseases are entered into the program to enable quarantine of livestock where applicable. Blood samples were also collected from 8969 sows on 2984 pig farms from 2016–2022 to measure anti-PEDV antibody titers. Every year (in August and September), blood was collected from three sows per pig farm and serum neutralizing antibody titers were measured. The purpose of investigating sow antibody titer was to allow pig farms to prepare for the damage to suckling pigs caused by diarrhea by measuring the PED antibody level of sows in the autumn, because the occurrence rate of PED is particularly high in the winter.

### 2.2. RT-PCR for PEDV

Total RNA was extracted from viral supernatants using the RNeasy mini kit (Qiagen, Cat. No. 74104, Maryland, MD, USA) and used to synthesize cDNA according to the protocol provided in the HelixCript (NanoHelix, Daejeon, Republic of Korea) kit. PCR was performed using AccuPower ProFi Taq PCR premix (Bioneer, Daejeon, Republic of Korea). Primers used were designed in sets of four to cover the complete PEDV S gene [27]. The PCR conditions were as follows: 5 min at 95 °C (initial denaturation), followed by 35 cycles of 20 s at 95 °C (denaturation), 30 s at 57 °C (annealing), 1 min at 68 °C (extension), and 5 min at 68 °C (final extension). The amplified PCR product was electrophoresed on a 1.0% agarose gel (Inclone Biotech Cat. No. IN4001-0500, Seongnam, Republic of Korea) containing ethidium bromide and observed using an ultraviolet image analyzer. The PCR products were then sequenced on an ABI Prism 3730xi DNA sequencer.

### 2.3. Phylogenetic Analysis of PEDV

The complete S gene sequence data of 211 PEDVs comprising the 140 strains examined in this study were analyzed alongside 71 worldwide reference strains obtained from the NCBI GenBank database. Multiple nucleotide sequence alignments were carried out using the Clustal X alignment program. The nucleotide sequences of the complete S gene were analyzed phylogenetically using the maximum-likelihood (ML) method, the Tamura–Nei model, and bootstrap analysis (*n* = 1000) in MEGA X software (with default parameters) [28]. The ML tree was based on the rates among sites (Gamma-distributed rate of invariant sites (G + I)) and the ML heuristic (nearest-neighbor-interchange (NNI)) methods. To categorize the genotypes of South Korean PEDV strains identified from 1997–2022, an ML tree of 340 South Korean PEDV strains, including the 140 strains in this study and 200 Korean reference strains in GenBank, was also constructed.

### 2.4. Changes in Population Size, Substitution Rates, and Recombination of PEDV

To explore the evolutionary history of the 140 PEDVs, substitution rates, time to the most recent common ancestor (tMRCA), and changes in population size were [29] co-estimated using the Bayesian coalescent approach in BEAST v1.8.4. The resulting convergence was analyzed using Tracer 1.6 [30], and statistical uncertainties were summarized as 95% highest posterior density (HPD) intervals. Trees were summarized as maximum clade credibility (MCC) trees using TreeAnnotator 1.7.5 [31], and visualized using Figtree 1.4.2 [32]. The change in effective population size over the time was traced using Bayesian skyline plot (BSP) analysis [33]. The complete nucleotide sequences of the S gene of the 140 PEDV were screened for the presence of recombination events using the RDP 3.0b41 package [34] with the default parameters. This package comprises these six different recombination detection programs: RDP, GENECONV, MaxChi, BOOTSCAN, Chimeara, and SiScan. This screening was performed only for putative recombinant sites detected by at least three out of the six methods to prevent detection of false-positive recombinants.

### 2.5. Serum Neutralization Antibody Tests 

Serum samples from sows were heat inactivated at 56 °C for 30 min and stored at −20 °C until use. Initially, serum samples were diluted 2-fold to detect PED-specific neutralizing antibodies. Briefly, an equal volume of virulent PEDV (SGP-M1 strain; 200TCID_50_/mL) was added to each sow serum sample in 96 well plates for 1 h. The reaction solution (pig serum/virulent PEDV) was then inoculated into a 96-well plate containing Vero cells and placed in a 37 °C incubator. Three days later, plates were checked for cytopathogenic effects (CPE) to determine the PED antibody titer. In cells infected with the SGP-M1 strain, characteristic cytopathic effect (CPE) patterns were observed including atypical syncytium formation and scattered cell nuclei. Serum neutralization titers were expressed as the reciprocal of the highest serum dilution that inhibited CPE.

### 2.6. Statistical Analysis

All statistical analyses were performed using GraphPad Prism 6. Statistical significance was evaluated by one-way ANOVA, and * *p* < 0.05 was considered significant.

## 3. Results

### 3.1. PEDV Incidence over 10 Years 

From 2013–2022, the KAHIS system listed the official number of farms with reported PED outbreaks in South Korea as 1131. The incidence was notably high in the winter–spring season (November to May of the following year) (Figure 1 and Table 1). Over the previous decade, the highest incidence of PED was in Jeju (JJ; 28.1%, 318/1131), followed by Chungnam (CN; 20.9%, 236/1131) and Jeonbuk (JB; 16.1%, 182/1131) (Table 1). During this period, South Korea experienced the following four major PED epidemics (Figure 1A,B): from November 2013 to May 2014; from November 2017 to May 2018; from November 2018 to May 2019; and from November 2021 to May 2022 (Figure 1A). During the four PED epidemics, pig farms in JJ, CN, and JB suffered numerous outbreaks (Figure 1B and Table 1). All 140 diarrheal pig samples obtained by nine local government livestock control agencies were positive for PEDV (Table 1). Of the 140 PEDVs, 28 were obtained in 2015, 26 in 2014, 22 in 2016, and 19 in 2022 (Table 1); of these 55 cases were in JJ, 27 in CN, and 26 in JB (Table 1).

### 3.2. ML Trees for the Complete S Gene Sequences

The ML tree revealed that 139 out of the 140 PEDVs analyzed in this study belong to the G2b genotype, while the remaining 1 belongs to the G2a genotype (Figure 2). In addition, PEDVs detected in South Korea between 1997 and 2012, which were also used as reference strains, belong to the G2a type (MF3809, KNU0905, KDJN12JU, NJ02, GW06CH, and AD03) or the G1 type (GN05DJ, CJ98, KUPE21, KNU-0801, Spk1, Chinju99, SM98, Virulent DR13, KPEDV-9, and Attenuated DR13) (Figure 2). ML tree analysis revealed that the 139 domestic PEDVs in this study belonging to the G2b type included strains from the United States (Iowa16465, OK10240-8, Colorado2013, IA1, Minnesota188, and PC273-O), Japan (14JM-311 and OKN-1), Canada (Quebec334), Thailand (P1915-NPF-071511A), Taiwan (PT-P5), Ukraine (Poltava01), and Mexico (MEX104 and MEX-QRO) (Figure 2). Two of the 139 strains (K17GG1 and K17GB3) were the S INDEL type and harbored a deletion within the S1 region of the S gene. These S INDEL-type strains belong to the same group as strains from European countries (Italy, Slovakia, France, Hungary, Germany, and Belgium), the United States, and Japan (Figure 2). Interestingly, 19 Korean PEDVs isolated in 2022 clustered with the PC273-O strain isolated in the USA and 4 older strains isolated in South Korea (K18JB1, K21GB1, K21GB2, and K21GB3) (Figure 2).

### 3.3. Changes in the Genotype of South Korean PEDVs by Year

Genotyping of the entire S gene of 340 PEDV strains (200 NCBI GenBank reference strains and the 140 strains in this study) detected in Korea from 1997–2022 (Figure 3) revealed that PEDVs identified in South Korea before 2000 were predominantly of the G1 type, whereas those identified from 2000–2012 were mainly of the G2a type (Figure 3). The G2b type, which was introduced into South Korea in late 2013, is the main strain found on pig farms to date (Figure 3). The S INDEL type, which belongs to the G2b type, was first detected in 2014, but at a low incidence, and was detected rarely between 2017 and 2019 (Figure 3). Also, the incidence of the G1 type was very low 2012–2014 (Figure 3).

### 3.4. Nucleotide Deletions and Insertion in the S Gene

Nucleotide sequence analysis of the complete S genes of 140 Korean PEDVs revealed that 14 harbored deletions and/or insertions. Two of the 14 (K17GG1 and K17GB3) were of the S INDEL type reported in the USA and Japan (Figure 4). Each of these two S INDEL types harbored deletions (Δ1, Δ11, Δ3, and Δ6) at four nucleotide sequence positions (167, 176–186, 413–415, and 469–474 or 479–484) that are located in the S1 domain, specifically in the S1-NTD region that is closely related to antigenicity (Figure 4B). Six (K15JJ2, K15GB1, K15CN5, K15CN6, K17GB1, and HS) of the 14 strains harbored nt deletions in the S1-NTD region, with the number of deletions being 3, 6, or 9 (Figure 4B). In addition, strain K15JJ1 harbored a 99 nt deletion in the S2 and HR1 regions (Figure 4B). Two strains (K16CN6 and SGP-M1) harbored a 3 nt and a 2 nt deletion in the S2 region, respectively (Figure 4B). Interestingly, three PEDV strains (K17JJ3, K17JJ4, and K17JJ7) had a 12 nt sequence inserted into the S1 region (Figure 4B).

### 3.5. PEDV Antibody Titers by Year

PEDV neutralizing antibody titers in sows were examined annually from 2016–2022. A total of 8969 sows from 2984 farms were tested over the 10-year period, and the mean neutralizing antibody titer was 3.78 ± 1.12 log_2_-fold (Table 2). The highest titer in sows was 4.96 ± 1.47 log_2_-fold in 2019, and the lowest was 1.39 ± 0.40 log_2_-fold in 2016 (Table 2). Notably, neutralizing antibody titers in 2021 were low (2.66 ± 0.78 log_2_-fold), which may have contributed to the 2022 PED epidemic (Table 2). In 2016, 64.4% of sows were negative for antibodies, whereas 31.4% were negative in 2021.

### 3.6. Genotype of PEDV According to Region

MCC tree analysis of the 140 PEDVs detected in each region in each year was performed using the BEAST program to investigate the tMRCA. The 140 PEDVs were largely divided into three clusters, with PEDVs from JJ forming an almost independent cluster B (Figure 5). Cluster A consisted primarily of PEDVs from the CN region, along with some PEDVs from other regions (Figure 5). Cluster C was dominated by PEDVs collected from the JB region in 2022, along with some PEDVs from other regions collected before 2022 (Figure 5).

### 3.7. Substitution Rate and Recombination of Spike Gene 

The national substitution rate (substitutions/site/year) for the complete spike genes of Korean PEDV detected from 2013–2022 was 1.2556 × 10^−3^ (95% highest posterior density (HPD) lower: 8.6827 × 10^−4^; 95% HPD upper: 1.5997 × 10^−3^) (Table 3). The substitution rates for the three hotspot regions with the most PEDV outbreaks were 1.5952 × 10^−3^ (1.2681 × 10^−3^, 1.9121 × 10^−3^) in CN, 1.8065 × 10^−3^ (1.4644 × 10^−3^, 2.1807 × 10^−3^) in JB, and 1.5113 × 10^−3^ (1.1315 × 10^−3^, 1.9121 × 10^−3^) in JJ (Table 3). The substitution mean rates in these three regions were slightly higher than the overall South Korea substitution rate (1.2556 × 10^−3^) (Table 3). The complete nucleotide sequences of the S gene of 140 PEDVs detected in South Korea were analyzed for recombination potential using the RDP 3.0b41 package. However, no putative recombination sites were detected using RDP, GENECONV, MaxChi, BOOTSCAN, Chimeara, or SiScan recombination detection programs.

### 3.8. Effective Population Size by Region 

The effective population size (number of PEDV-positive farms) was measured using BSPs, and the incidence trend was measured according to region (Figure 6). In the case of JJ, PED outbreaks peaked in 2014 and have continued to occur since then, albeit with a somewhat decreasing pattern (Figure 6A). However, in the case of CN, a high incidence was observed from the second half of 2018, which continued to 2022 (Figure 6B). A pattern of continuous occurrence has been observed in JB since 2014 (Figure 6C).

## 4. Discussion

The non-INDEL strain of PEDV (G2b), which emerged in 2013, is highly pathogenic and has caused significant damage to suckling pigs; outbreaks have since been confirmed in Canada, Mexico, Japan, Korea, Thailand, Taiwan, and the Philippines [27,35,36,37,38,39]. In addition, the S INDEL, a new type of PEDV that originated in the United States and is less pathogenic than the non-INDEL form, has since spread to European countries such as Germany [40], Belgium [41], France [42], Italy [43,44], Austria [45], Portugal [46], Slovenia [47], and Hungary [48]. The highly pathogenic non-INDEL type has been identified only in Ukraine among European countries [49]; however, there is concern about whether this type will spread to other European countries. Like the United States [50], the non-INDEL and S INDEL types (OH851 strain) have been reported in Korea [51] and Japan [52]. Here, we found that 139 of 140 PEDV strains belonged to the G2b type, although only 2 (K17GG1 and K17GB3) were similar to the OH851 strain of American PEDV (Figure 4). However, 12 PEDV strains did not harbor deletion of the same nucleotide sequences as the PEDV OH851 strain; rather, deletions and insertions were detected in other regions (S1-NTD or S2) of the S gene. The PEDV S protein is responsible for receptor binding and viral entry, and, as such, it determines host range and cell tropism [53,54]. In addition, neutralizing epitopes are present in the S protein, making it the primary target of PEDV vaccines [55,56]. Since neutralizing antibodies play an important role in preventing and controlling viral infections, it is important to analyze changes in the amino acid sequence of the viral proteins. Four neutralizing epitopes (COE, 2C10, SS2, and SS6) have been identified in the PEDV S protein [57,58,59]. However, evolutionary changes in the S gene regions responsible for immunogenicity and pathogenicity are expected to emerge continually. 

According to the analysis of the genotypes of previously published Korean reference PEDVs, the G1 and G2a types existed between 1997 and 1999, the G2a type was predominant between 2000 and 2012, and only G2b type was identified after 2015. The recent pattern of PED outbreaks in Korea has been dominated by non-INDEL strains (G2b genotype), and the incidence of S INDEL strains (G2b genotype) has been very low; indeed, none have been confirmed since 2020. In a previous study, all global PEDV isolates (except one unclassified isolate) were classified into six groups [60]. Among these, groups 1 to 5 comprise pandemic viruses, while group 6 comprises classical isolates. Within the pandemic clade, groups 1 and 2 originated in North America, whereas groups 3 to 5 originated in Asia, with group 2 originating from S INDEL isolates. The authors report no clear link between temporal or geographic origin and heterogeneity of PEDVs within each group [60]. Here, we confirmed that groups 3–5 belong to the G2a type, group 1 is classified as non-INDEL and belongs to G2b, group 2 is classified as an S INDEL and belongs to G2b, and classical group 6 viruses belong to G1. Analysis of spatiotemporal spread (Bayesian phylogeographic analysis) of S protein-coding genes confirmed that the G2b-type PEDV detected in South Korea since late 2013 originated in China and the United States [61]. Analysis of PEDV S genes detected in South Korea from 1998–2013 suggested that conservative substitutions, suggestive of positive selection, were more common in the S gene than in the PEDV genome as a whole [27]. A previous study analyzed 138 published genome sequences using Bayesian coalescent analyses, as well as Bayesian inferencing and ML methods [60]. The results suggest that the PEDV virus evolved at a rate of 3.38 × 10^−4^ (95% HPD: 2.75 × 10^−4^–6.72 × 10^−4^) substitutions/site/year, and the tMRCA emerged 75.9 (95% HPD: 37.01–126.65) years ago [60]. However, the rate of PEDV evolution in South Korea (1.2556 × 10^−3^, substitutions/site/year) is slightly faster than the global rate. Almost all RNA viruses undergo 10^−2^–10^−5^ nucleotide substitutions per site per year [62]. This rapid rate of evolution of RNA viruses, including PEDV, generally arises from several factors, including short generation time, small genome size, rapid mutation, and lack of polymerase proofreading [63]. As a result, RNA viruses spread rapidly into new environments and hosts, thereby increasing the adaptability, viability, and fitness of the viral population [64].

The new G1b variant arose from spontaneous recombination between the pandemic G2b virus and a small number of G1b viruses circulating in South Korea, resulting in two G1b-type recombinant viruses (KNU-1808 and KNU-1909) [26]. Another study reported that Korean G2b-type strain J3142, presumably generated by recombination, harbored a potential recombination breakpoint (376–2143 nt) in the S1 gene between KNU1303 Korean strain G2a (KJ451046) and 45RWVCF0712 Thai strain G2b (KF724935) [61]. However, we found no evidence of S gene recombination in any of the 140 Korean PEDV strains tested using the recombination detection program (RDP).

In a previous study in Korea, a neutralization test was conducted by crossing the PEDV antibody titers against suckling piglets, gilt, and sow with the G2b-based BM3 strain and the G1-based DR13 strain [61]. They suggested that the pandemic G2b virus was partially neutralized by antibodies induced by the G1-based PED vaccine virus [61]. Therefore, since 2016, annual neutralization tests have been conducted using serum samples from domestic sows and the G2b strain vaccination is encouraged when anti-PEDV antibody titers are low. In 2017, the antibody titer was 4.82 ± 1.54 log_2_-fold, whereas it was 3.94 ± 1.15 log_2_-fold in 2018; however, this did not prevent epidemic PED outbreaks. In addition, prior to the epidemic of 2022, the antibody titer in sows in 2021 was low (2.66 ± 0.78 log_2_-fold), resulting in great damage when the epidemic occurred (Table 2). The low sow antibody rate nationwide in 2021 is believed to be the reason why pig farms avoided vaccination, because the number of PED cases in Korea was very low from the second half of 2019 to the first half of 2021. While IgG antibodies play an important role in preventing PEDV infection, systemic and mucosal immune responses, including secretion of virus-specific IgA, are essential [65,66,67]. Therefore, to predict PEDV epidemics, it is recommended that neutralizing antibody and IgA antibody titers in sows are measured simultaneously.

The purpose of the genetic cluster analysis in this study was to estimate how PED viruses are clustered regionally and the evolution rate of PEDV by region. Since the Jeju (JJ) area, which belongs to cluster B, is an island and is separated from the mainland, it is assumed that PEDV circulation occurred and evolved independently rather than being frequently introduced from the mainland (Figure 5). Conversely, clusters A and B areas are on the mainland, so it is presumed that PEDV influx between regions on the mainland was frequent and that genetic mutations occurred more frequently as a result.

In this study, we found that the effective population size of PEDVs from 2013–2022, as assessed by BSP analysis, showed a decreasing incidence in the JJ region toward 2022, but a steady incidence in the CN and JB regions (Figure 6). The BSP in the Bayesian MCC phylogenetic tree derived from the complete coding genome sequences of 138 global PEDVs showed that the effective population size remained consistent, except during a short period around 2012 [60]. Recently, a major factor contributing to transmission of PEDV in China has been geographic spread due to the transportation of live pigs [68,69].

A major contributor to the spread of PEDVs in South Korea is potential transmission by pig transport vehicles entering and leaving slaughterhouses (samples have tested PCR-positive; unpublished data). In South Korea, slaughter is not designated to take place in specific slaughterhouses; rather, it can take place anywhere in the country, which can lead to cross-regional spread. A major problem with the spread of PEDV is that information about the outbreak farm is not communicated to other pig farms and slaughterhouses. In addition, it is presumed that a secondary reason for the spread of PEDV in South Korea is that farms do not vaccinate their pigs or use only G1-type vaccines, which provide only partial protection. Therefore, to prevent the spread of PEDV in South Korea, thorough disinfection of pig transport vehicles and movement restrictions must be performed in parallel, and farms must use appropriate vaccines and share PED occurrence information between farms. Future domestic PEDV outbreaks are expected to occur periodically (at approximately 3-year intervals), and will continue to be dominated by highly pathogenic G2b-type non-INDEL strains.

## 5. Conclusions

Analysis of the complete S genes of 140 PEDVs prevalent in Korea from 2013–2022 identified 137 highly pathogenic non-INDEL strains (G2b genotype), 2 less pathogenic S INDEL strains (G2b genotype), and 1 strain belonging to the G2a genotype. In particular, among the three hot spot regions in which PED epidemics are causing great damage, PEDV in the Jeju region appears to be undergoing independent genetic evolution, and the incidence of PED is predicted to fall in the future. Two mainland regions (CN and JB) harbor mixed clusters containing PEDVs from other regions, and the incidence of PED in these regions is expected to remain the same. In South Korea, the highly pathogenic non-INDEL type G2b will continue to be the dominant strain of PEDV.

## Figures and Tables

**Figure 1 viruses-15-02165-f001:**
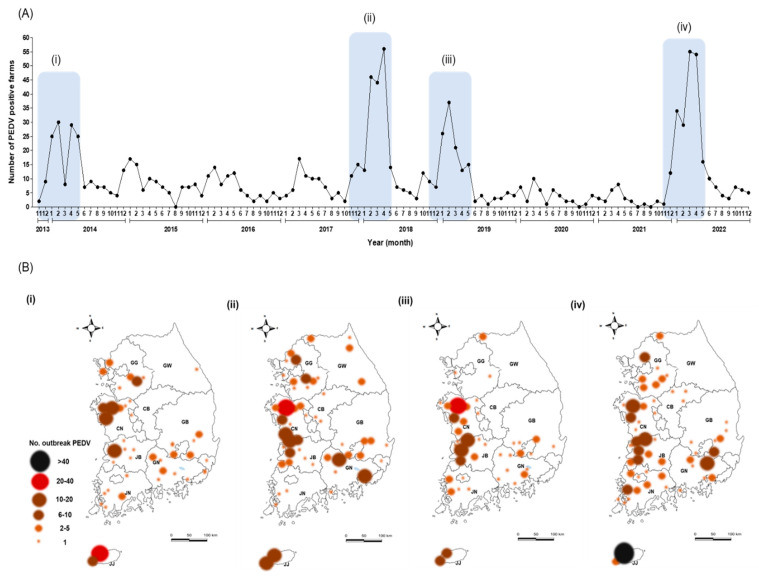
PED occurrence from 2013–2022, based on information from the KAHIS (Korean Animal Health Investigation System, https://www.kahis.go.kr, accessed on 3 April 2023). The major epidemic PED outbreaks (i–iv) are denoted by year and month (**A**), and the pig farms involved in the outbreak area are marked on the map (**B**). GG: Gyeonggi; GW: Gangwon; CN: Chungnam; CB: Chungbuk; GN: Gyeongnam; GB, Gyeongbuk; JN: Jeonnam; JB: Jeonbuk; JJ: Jeju Island.

**Figure 2 viruses-15-02165-f002:**
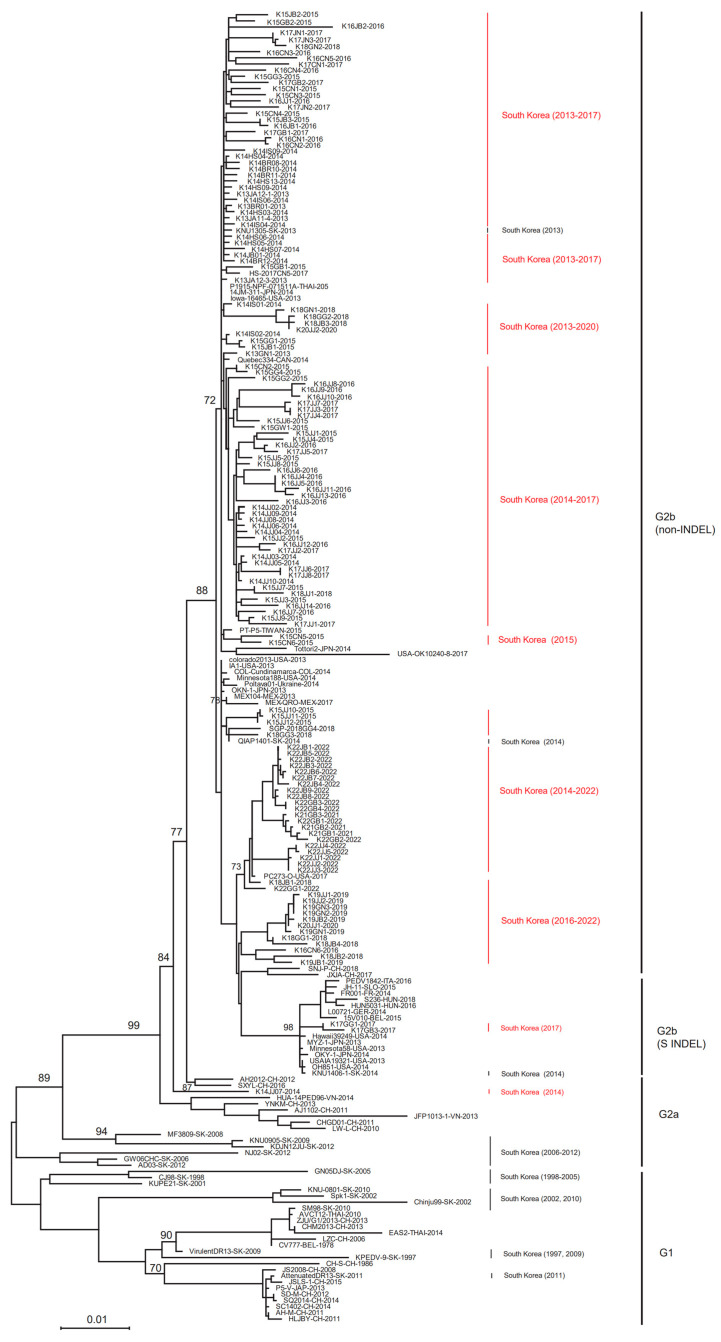
Phylogenetic tree analysis of the complete spike genes of the 211 PEDVs, comprising the 140 PEDV analyzed in this study and 71 global PEDV reference sequences. Red font (South Korea) denotes PEDVs detected in this study, and black font (South Korea) denotes reference strains obtained from GenBank.

**Figure 3 viruses-15-02165-f003:**
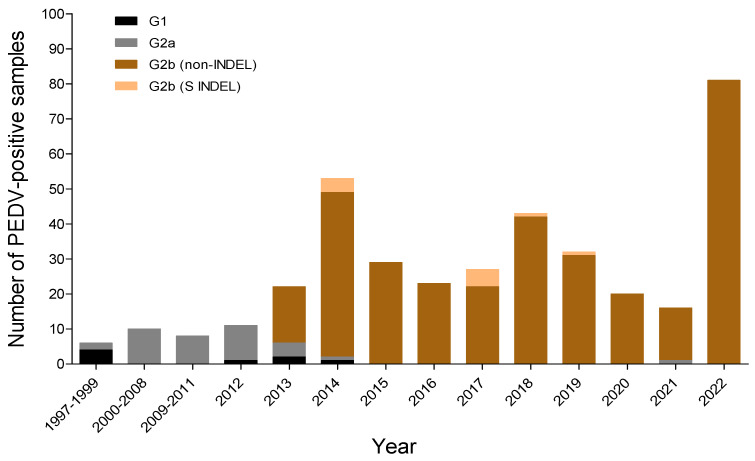
PEDV genotypes detected in South Korea from 1997–2022. ML analysis of 340 Korean PEDVs (the 140 PEDVs used in this study and 200 Korean reference PEDVs registered in GenBank) was conducted. The four genotypes are shown as black (G1), gray (G2a), light brown (G2b, non-INDEL), and peach (G2b, S INDEL) bars.

**Figure 4 viruses-15-02165-f004:**
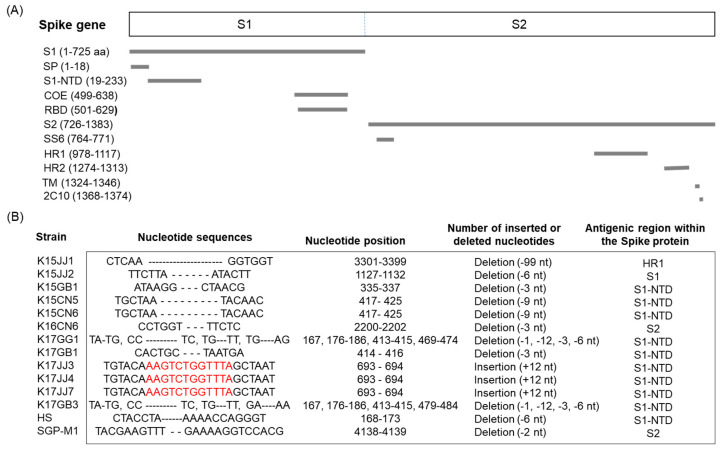
Deletions and insertions in the spike gene of Korean PEDV strains. Antigenic sites within the spike protein (**A**), and the 14 Korean PEDV strains harboring insertions and deletions (**B**). Deletions are marked by dashes (–), and insertions by red letters.

**Figure 5 viruses-15-02165-f005:**
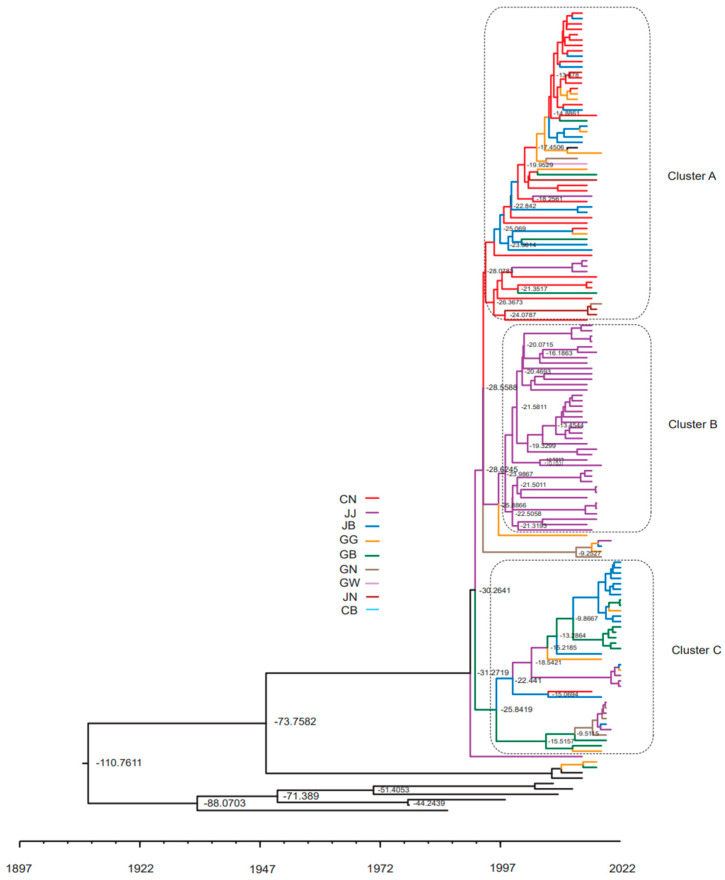
Phylogenetic tree of the complete spike gene sequences of Korean PEDV collected in nine regions. MCC tree analysis of the 140 Korean PEDV strains detected in this study from 2013–2022 was conducted using the BEAST program. Each of the nine regions is marked by a different colored line. Bar = expected year. GG: Gyeonggi; GW: Gangwon; CN: Chungnam; CB: Chungbuk; GN: Gyeongnam; GB, Gyeongbuk; JN: Jeonnam; JB: Jeonbuk; JJ: Jeju Island.

**Figure 6 viruses-15-02165-f006:**
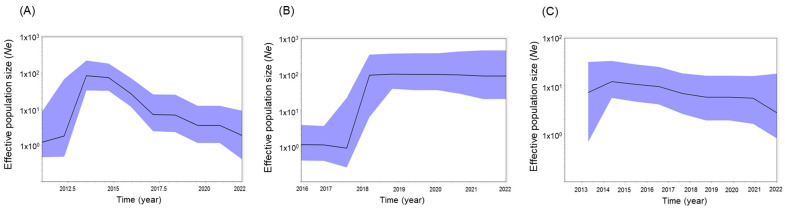
Effective population size of PEDV from 2013–2022, presented as Bayesian skyline plots, in Jeju (**A**), Chungnam (**B**), and Jeonbuk (**C**).

**Table 1 viruses-15-02165-t001:** Number of PEDV samples collected and the number of PEDV occurrences in South Korean domestic pigs from 2013–2022.

Year	No. PED Outbreaks on Pig Farms * (No. of PEDV Samples Collected in This Study)									
	GG **	GW	CN	CB	GN	GB	JN	JB	JJ	Total
2013	2 (2)	0	5 (2)	0	3 (1)	2	0	0	0	12 (5)
2014	22	3	47 (11)	0	12	13	8	21 (6)	43 (9)	169 (26)
2015	11 (4)	2 (1)	15 (6)	0	3	6 (2)	4	6 (3)	48 (12)	95 (28)
2016	12	0	11 (6)	0	5	5	1	7 (2)	41 (14)	82 (22)
2017	11 (1)	1	13 (2)	0	11	10 (3)	6 (1)	2	47 (10)	101 (17)
2018	29 (4)	10	62	4	22 (2)	17	3	41 (4)	34 (1)	222 (11)
2019	1	4	42	2	9 (3)	13	9	44 (2)	10 (2)	134 (7)
2020	0	0	5	1	1	7	10	10	11 (2)	45 (2)
2021	6	0	0	1	5	11 (3)	2	4	11	40 (3)
2022	16 (1)	2	36	3	16	14 (4)	24	47 (9)	73 (5)	235 (19)
Total	110 (12)	22 (1)	236 (27)	11	87 (6)	98 (12)	67 (1)	182 (26)	318 (55)	1131 (140)

* Number of PED outbreaks reported to the Korean Animal Health Investigation System (KAHIS) system. ** GG: Gyeonggi; GW: Gangwon; CN: Chungnam; CB: Chungbuk; GN: Gyeongnam; GB, Gyeongbuk; JN: Jeonnam; JB: Jeonbuk; JJ: Jeju Island.

**Table 2 viruses-15-02165-t002:** Serum-neutralizing antibody titers in Korean sows from 2016 to 2022.

Year	2016	2017	2018	2019	2020	2021	2022	Total
No. of pig farms	566	405	443	341	502	415	312	2984
No. of sows	1713	1215	1329	1023	1506	1245	938	8969
SN antibodies ^a^	1.39 ± 0.40	4.82 ± 1.54	3.94 ± 1.15	4.96 ± 1.47	4.89 ± 1.44	2.66 ± 0.78	3.77 ± 1.09	3.78 ± 1.12

^a^ Serum neutralizing (SN) antibody titer: (average log_2_-fold) ± SD.

**Table 3 viruses-15-02165-t003:** Substitution rates in the PEDV spike gene between 2013 and 2022.

PEDV Genotype	Region	Substitution Rate (Substitutions/Site/Year)		
		Mean	95% HPD Lower	95% HPD Upper
G2b	SK *	1.2556 × 10^−3^	8.6827 × 10^−4^	1.5997 × 10^−3^
	CN	1.5952 × 10^−3^	1.2681 × 10^−3^	1.9121 × 10^−3^
	JB	1.8065 × 10^−3^	1.4644 × 10^−3^	2.1807 × 10^−3^
	JJ	1.5113 × 10^−3^	1.1315 × 10^−3^	1.9121 × 10^−3^

* SK, South Korea; CN, Chungnam; JB, Jeonbuk; JJ, Jeju Island.

## Data Availability

The nucleotide sequences of the complete S gene of the 140 PEDV strains obtained in this study were submitted to the GenBank database under accession numbers (KJ539151-KJ539154, KM924403-KM924428, OR045906-OR045907, OR604147-OR604253, and OR652060).

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
