# Peer review of "Spike Gene Analysis and Prevalence of Porcine Epidemic Diarrhea Virus from Pigs in South Korea: 2013–2022"

_viruses, 2023, doi:10.3390/v15112165_

Round 1

Reviewer 1 Report

Comments and Suggestions for Authors

The article “Spike gene analysis and prevalence of porcine epidemic 2 diarrhea virus from pigs in South Korea: 2013–2022” by Gyu-Nam Park and colleagues examines the distribution of spike gene variants in South Korea. The authors have done serious work, including a thorough bioinformatics analysis. This is meticulous and informative research. The manuscript is well organized and the illustrations are well done and useful. Analysis of evolutionary history and recombination events can contribute to PEDV research. Overall, the manuscript is well written and includes a clear results section and a brief discussion.

The reviewer's main concern is the lack of a structural description of spikes. Perhaps modern artificial intelligence methods such as AlphaFold will allow us to easily predict the structure and effect of mutations with high enough accuracy to illustrate the possible consequences of mutations. And there are a couple of small notes:

Line 99 – Have you found the best substitution model? Are you sure that models other than G+I do not provide significant advantages?

Line 310. “However, the development rate of PEDV in South Korea (1.2556 × 10−3, 310 replacements/site/year) is slightly higher than the global rate.” - Do you have any idea why this is happening?

Author Response

Reviewer 1

The article “Spike gene analysis and prevalence of porcine epidemic 2 diarrhea virus from pigs in South Korea: 2013–2022” by Gyu-Nam Park and colleagues examines the distribution of spike gene variants in South Korea. The authors have done serious work, including a thorough bioinformatics analysis. This is meticulous and informative research. The manuscript is well organized and the illustrations are well done and useful. Analysis of evolutionary history and recombination events can contribute to PEDV research. Overall, the manuscript is well written and includes a clear results section and a brief discussion.

Comment 1: The reviewer's main concern is the lack of a structural description of spikes. Perhaps modern artificial intelligence methods such as Alpha Fold will allow us to easily predict the structure and effect of mutations with high enough accuracy to illustrate the possible consequences of mutations. And there are a couple of small notes:

Answer: No member of our team feels sufficiently competent to analyze spike protein structure, and there are no experts in our laboratory who can handle protein structure programs. Therefore, we apologize for not being able to perform the protein tertiary structure analysis requested by the reviewer.

Comment 2: Line 99 – Have you found the best substitution model? Are you sure that models other than G+I do not provide significant advantages?

 Answer: The mega program used in this manuscript provides only four rates among sites, which are uniform rates, gamma distributed (G), invariant sites (I), and gamma distributed with invariant sites (G+I). Therefore, the best substitution model cannot be obtained using the Mega program, and even if a best model could be selected from another program, it cannot be applied to the Mega program. Therefore, we used the most common and universal gamma distribution (G+I).

Comment 3: Line 310. “However, the development rate of PEDV in South Korea (1.2556 × 10−3, 310 replacements/site/year) is slightly higher than the global rate.” - Do you have any idea why this is happening?

Answer: There are two main reasons why PEDVs in South Korea evolve slightly faster than PEDVs elsewhere in the world. First, South Korea has many pig farms that are concentrated in close proximity in a small area and there continues to be a high number of PED cases per year within this region. Therefore, the rate of genetic mutation of PEDV is expected to be faster. Second, as live PEDV vaccines are used more than in other countries, genetic mutations are expected due to recombination between PEDV vaccine strains and field PEDVs.

Reviewer 2 Report

Comments and Suggestions for Authors

Porcine epidemic diarrhea virus (PEDV), belonged to Alphacoronavirus, results in acute diarrhea, dehydration and high mortality in piglets. In this study, Park and colleagues monitor the serum-neutralizing antibody titer, geological and genomic changes in PEDV from 2013-2022 in South Korea. This study is important and appreciated. However, it is hard to see the importance and the implications of this study, and also numerous major concerns must be addressed.

Major concerns:

1.    Line 62: it is recommended to have more information about mutations found in S INDEL strains.

2.    Line 73: What criteria did the authors specifically choose those 140 pig farms in this study?

3.    Line 81-82: Any references to support the claim?

4.    Line 94 and 102: which reference strains were used? In line 94, 71 ref strains were mentioned while 200 references strains were mentioned in line 102. Are they different? The information is quite confusing.

5.     Line 125: CPE was mentioned. It is highly recommended to include examples in this manuscript.

6.    Fig 1B: the data is not clearly presented. Higher resolution is required.

7.    Table 1: what does the numbers in blankets mean?

8.    Fig 2: it is highly recommended to label the reference sequences.

9.    Fig 3: the reason to specifically present the INDEL strains was not clear. Given that, INDEL is also belonged to G2b strain.  

10. Section 3.4: 14 PEDVs were stated but only 13 virus strains were described. What characteristics does the remaining one have? Why are the strains with deletion or insertion specifically described? How about other 126 PEDV strains?

11. Table 2: What does ‘SN’ mean?

12. Fig 5: Full name of the place should be used.

13. Section 3.6: PEDVs from JJ were found in cluster B. Any implications or explanations of this observation? What is the purpose to perform this analysis? Further discussion is required.

14. Table 3: What does “HPD lower/higher” mean?

15. Line 249-250: The authors mentioned that “The substitution rates in these three regions were slightly lower than 249 the overall South Korea substitution rate.”. Statistical analysis is required to tell if there are any differences.

16. It is highly recommended to define “the effective population size” in the analysis.

17. It is highly recommended to quote the figures or tables in Discussion.

18. In line 304-305, the authors mentioned that the evolution of Korean strains has been slow, while they also stated that the rate of PEDV evolution in South Korea is slightly faster than the global rate in line 310-311. The idea is confusing, and references are required.

19. Line 324-325: references or results are required to support the statement.  

20. In this study, the evolution of PEDV genome and the change in the antibody titer from pig serum were analyzed. Are there any relationships? Further discussion is highly recommended.

21. The neutralizing antibody titers in Korea should be compared to those in other findings. It seemed that the titers were relatively lower than previous findings. Any explanations? In Table 2, a sudden drop of the titer was found in 2021. Why? Are there any relationships with the circulating stains in 2021?

22. Any potential measures to prevent the virus spreading could be included in Discussion.

Comments on the Quality of English Language

Proof-reading has to be carried out carefully.

Author Response

Reviewer 2

Porcine epidemic diarrhea virus (PEDV), belonged to Alphacoronavirus, results in acute diarrhea, dehydration and high mortality in piglets. In this study, Park and colleagues monitor the serum-neutralizing antibody titer, geological and genomic changes in PEDV from 2013-2022 in South Korea. This study is important and appreciated. However, it is hard to see the importance and the implications of this study, and also numerous major concerns must be addressed.

Major concerns:

Comment 1: Line 62: it is recommended to have more information about mutations found in S INDEL strains.

Answer: We revised the sentence (Revised manuscript lines: 62-66).

Although it was not the same as the S INDEL strain that is prevalent in the United States, cases of INDEL strains with genetic mutations have also been reported in South Korea [24,25], including one strain with 15 nucleotides inserted in the S1 gene region (KOR/KNU-1601/2016) and two strains with six nucleotides deleted in the S2 gene region (KOR/KNU-1829/2018 and KOR/KNU- 1830/2018) [24,25]. (Revised manuscript lines: 62-66).

Comment 2: Line 73: What criteria did the authors specifically choose those 140 pig farms in this study?

Answer: PEDV testing on diarrhea specimens from pig farms is conducted by local government laboratories across the country. Additionally, local governments input details of PEDV positive cases into the KAHIS system. We, the central government, request local government labs across the country to share samples so that we can analyze the positive samples. However, because local government laboratories across the country discard most PEDV-positive samples after testing, only 140 positive samples were shared with us for analysis.

Comment 3: Line 81-82: Any references to support the claim?

Answer: We have revised the sentence as follows “The purpose of investigating sow antibody titer was to allow pig farms to prepare for the damage caused by diarrhea to suckling piglets by measuring the PED antibody level of sows in the autumn because the occurrence rate of PED is particularly high in the winter” (Revised manuscript line 84-87).

Comment 4: Line 94 and 102: which reference strains were used? In line 94, 71 ref strains were mentioned while 200 references strains were mentioned in line 102. Are they different? The information is quite confusing.

Answer: The 71 reference strains mentioned in line 94 were representative of PEDVs in each country. However, the 200 reference strains in line 102 are a collection of all Korean PEDV strains registered in Genbank up until now. We revised the sentences (Revised manuscript lines: 100-102, 108-110). The complete S gene sequence data of 211 PEDVs comprising the 140 strains examined in this study were analyzed alongside 71 worldwide reference strains obtained from the NCBI GenBank database (Revised manuscript lines: 100-102). To categorize the genotypes of South Korean PEDV strains identified between 1997–2022, an ML tree of 340 South Korean PEDV strains, including the 140 strains in this study and the 200 Korean reference strains in GenBank, was also constructed (Revised manuscript lines: 108-110).

Comment 5: Line 125: CPE was mentioned. It is highly recommended to include examples in this manuscript.

Answer: We added the sentence “In cells infected with the SGP-M1 strain, characteristic cytopathic effect (CPE) patterns were observed including atypical syncytium formation and scattered cell nuclei” (Revised manuscript lines: 132-134).

Comment 6: Fig 1B: the data is not clearly presented. Higher resolution is required.

Answer: We replaced Fig 1B with a higher resolution one.

Comment 7: Table 1: what does the numbers in blankets mean?

Answer: As mentioned in parentheses in Table 1, it means “Number of PEDV samples collected in this study”.

Comment 8: Fig 2: it is highly recommended to label the reference sequences.

Answer: We revised the Figure 2 legend “Figure 2. Phylogenetic tree analysis of the complete spike genes of 211 PEDVs, comprising those of the 140 PEDVs analyzed in this study and the 71 global PEDV reference sequences.” (Revised Figure 2 legend lines: 188-191).

Comment 9: Fig 3: the reason to specifically present the INDEL strains was not clear. Given that, INDEL is also belonged to G2b strain.  

Answer: We revised Fig 3 and its legend (Revised Figure 3 and legend). The four genotypes are shown as black (G1), gray (G2a), light brown (G2b, non-INDEL), and peach (G2b, S INDEL) bars (Revised lines: 206-207).

Comment 10: Section 3.4: 14 PEDVs were stated but only 13 virus strains were described. What characteristics does the remaining one have? Why are the strains with deletion or insertion specifically described? How about other 126 PEDV strains?

Answer: We revised the sentences (Revised manuscript lines: 209-220), and the remaining 126 PEDV spikes gene sequences showed no insertions or deletions.

Nucleotide sequence analysis of the complete S genes of 140 Korean PEDVs revealed that 14 harbored deletions and/or insertions. Two of the 14 (K17GG1 and K17GB3) were of the S INDEL type reported in the USA and Japan (Figure 4). Each of these two S INDEL types harbored deletions (â–³1, â–³11, â–³3, and â–³6) at four nt sequence positions (167, 176-186, 413–415, and 469–474 or 479–484), which are located in the S1 domain, specifically in the S1-NTD region that is closely related to antigenicity (Figure 4B). Six (K15JJ2, K15GB1, K15CN5, K15CN6, K17GB1, and HS) of the 14 strains harbored nt deletions in the S1-NTD region, with the number of deletions being three, six or nine (Figure 4B). In addition, strain K15JJ1 harbored a 99 nt deletion in the S2 and HR1 regions (Figure 4B). Two strains (K16CN6 and SGP-M1) harbored a 3 nt and a 2nt deletion in the S2 region, respectively (Figure 4B). Interestingly, three PEDV strains (K17JJ3, K17JJ4, and K17JJ7) had a 12 nt sequence inserted into the S1 region (Figure 4B). (Revised manuscript lines: 209-220)

Comment 11: Table 2: What does ‘SN’ mean?

Answer: SN is an abbreviation of serum-neutralizing (SN) (Revised manuscript line: 234).

Comment 12: Fig 5: Full name of the place should be used.

Answer: We have added “GG: Gyeonggi; GW: Gangwon; CN: Chungnam; CB: Chungbuk; GN: Gyeongnam; GB, Gyeongbuk; JN: Jeonnam; JB: Jeonbuk; and JJ: Jeju island.” (Revised manuscript line: 248-249).

Comment 13: Section 3.6: PEDVs from JJ were found in cluster B. Any implications or explanations of this observation? What is the purpose to perform this analysis? Further discussion is required.

Answer: We added a discussion about this (Revised manuscript lines: 353-359).

“The purpose of the genetic cluster analysis in this study was to estimate how PED viruses are clustered regionally and the evolution rate of PEDV by region. Since the Jeju (JJ) area, which belongs to cluster B, is an island and is separated from the mainland, it is assumed that PEDV circulation occurred and evolved independently rather than being frequently introduced from the mainland (Figure 5). Conversely, clusters A and B areas are on the mainland, so it is presumed that PEDV influx between regions on the mainland was frequent and that genetic mutations occurred more frequently as a result.” (Revised manuscript lines: 353-359).

Comment 14: Table 3: What does “HPD lower/higher” mean?

Answer: HPD lower/higher means lower and higher levels of 95% highest posterior density [HPD] (Revised manuscript lines 252-253).

Comment 15: Line 249-250: The authors mentioned that “The substitution rates in these three regions were slightly lower than 249 the overall South Korea substitution rate.”. Statistical analysis is required to tell if there are any differences.

Answer: We revised the sentence “The substitution mean rates in these three regions were slightly lower than the overall South Korea substitution rate (1.2556 × 10-3) (Table 3) (Revised manuscript lines 256-257).

Comment 16: It is highly recommended to define “the effective population size” in the analysis.

Answer: We used the Bayesian skyline plot (BSP) method to estimate changes in effective population size over time and the increase and decrease in PEDV in Korea from 1997 to 2022. In other words, the change in effective population size refers to the number of positive pig farms infected with PEDV.

We revised the sentence “The effective population size (number of PEDV-positive farms) was measured using the Bayesian skyline plot (BSP) method, and the incidence trend was measured according to region (Figure 6)” (Revised manuscript lines: 266-267)

Comment 17: It is highly recommended to quote the figures or tables in Discussion.

Answer: We have cited tables and figures in the Discussion.

Comment 18: In line 304-305, the authors mentioned that the evolution of Korean strains has been slow, while they also stated that the rate of PEDV evolution in South Korea is slightly faster than the global rate in line 310-311. The idea is confusing, and references are required.

Answer: We removed the sentence (Revised manuscript lines: 316).

We removed “It suggested that the evolution of Korean strains has been slow [27].”

Comment 19: Line 324-325: references or results are required to support the statement.  

Answer: We have added sentences below (Revised manuscript lines: 250, 258-261, 335-336).

3.7. Substitution rate and recombination of spike gene (Revised manuscript lines: 250).

The complete nucleotide sequences of the S gene of 140 PEDVs detected in South Korea were analyzed for recombination potential using the RDP 3.0b41 package. However, no putative recombination sites were detected using RDP, GENECONV, MaxChi, BOOTSCAN, Chimeara, or SiScan recombination detection programs (data not shown). (Revised manuscript lines: 258-261).

However, we found no evidence of S gene recombination in any of the 140 Korean PEDV strains tested using the recombination detection program (RDP) (data not shown). (Revised manuscript lines: 335-336).

Comment 20: In this study, the evolution of PEDV genome and the change in the antibody titer from pig serum were analyzed. Are there any relationships? Further discussion is highly recommended.

Answer: We could not analyze the evolution of the PEDV genome or changes in pig serum antibodies because the following conditions were not met: (1) experiments must first be performed to analyze the evolution of the entire PEDV genome and (2) these viruses must be isolated and correlated with pig serum antibody titers. Therefore, I apologize that we cannot provide a satisfactory response to this comment.

Comment 21: The neutralizing antibody titers in Korea should be compared to those in other findings. It seemed that the titers were relatively lower than previous findings. Any explanations? In Table 2, a sudden drop of the titer was found in 2021. Why? Are there any relationships with the circulating stains in 2021?

Answer: There has been little research on PED-neutralizing antibody titer data in Korea, making comparative analysis difficult.

We added the sentence below (Revised manuscript lines: 346-349).

The low sow antibody rate nationwide in 2021 is believed to be the reason why pig farms avoided vaccination because the number of PED cases in Korea was very low from the second half of 2019 to the first half of 2021.

Comment 22: Any potential measures to prevent the virus spreading could be included in Discussion.

Answer: We added the sentences below (Revised manuscript lines: 372-378).

In addition, it is presumed that the second reason for the spread of PEDV in South Korea is that farms do not vaccinate their pigs or use only G1-type vaccines, which provide only partial protection. Therefore, to prevent the spread of PEDV in South Korea, thorough disinfection of pig transport vehicles and movement restrictions must be performed in parallel, and farms must use appropriate vaccines and share PED occurrence information between farms. (Revised manuscript lines: 372-378).

Reviewer 3 Report

Comments and Suggestions for Authors

The research manuscript titled "Spike gene analysis and prevalence of porcine epidemic diarrhea virus from pigs in South Korea: 2013–2022" by Park et al is an interesting work from South Korea. The authors conducted a nationwide analysis of spike gene of PEDV (n = 140) and concluded that 139 smples belonged to G2b genotype and only one sample belonged to the G2a genotype. They further reported that highly pathogenic non-INDEL type G2b genotype of PEDV will continue to be the dominant strain in South Korean pig populations. 

Overall, the research design is appropriate and the manuscript is well written. The research findings reported would be useful for the stakeholders in South Korea. 

This reviewer has only a few minor suggestions:

1. L84: RNeasy mini kit (Qiagen, USA) or RNeasy mini kit (Qiagen, Germany)? Also, please provide the catalogue number of the kit used. 

2. L90: Please also provide information on the agarose gel electrophoresis.

3. Please write a brief opening and a closing paragraph (after Figure 6) for results section. 

Author Response

Reviewer 3

The research manuscript titled "Spike gene analysis and prevalence of porcine epidemic diarrhea virus from pigs in South Korea: 2013–2022" by Park et al is an interesting work from South Korea. The authors conducted a nationwide analysis of spike gene of PEDV (n = 140) and concluded that 139 smples belonged to G2b genotype and only one sample belonged to the G2a genotype. They further reported that highly pathogenic non-INDEL type G2b genotype of PEDV will continue to be the dominant strain in South Korean pig populations. 

Overall, the research design is appropriate and the manuscript is well written. The research findings reported would be useful for the stakeholders in South Korea. 

This reviewer has only a few minor suggestions:

Comment 1: L84: RNeasy mini kit (Qiagen, USA) or RNeasy mini kit (Qiagen, Germany)? Also, please provide the catalogue number of the kit used.  

Answer: We added “Total RNA was extracted from viral supernatants using the RNeasy mini kit (Qiagen, Cat. No. 74104, USA)” (revised manuscript lines: 89-90).

Comment 2: L90: Please also provide information on the agarose gel electrophoresis.

Answer: We added “The amplified PCR product was electrophoresed on a 1.0% agarose gel (Inclone Biotech Cat. No. IN4001-0500, South Korea) containing ethidium bromide and observed using an ultraviolet image analyzer” (revised manuscript lines: 95-97).

Comment 3: Please write a brief opening and a closing paragraph (after Figure 6) for results section. 

Answer: We divided paragraph 3.7 (Substitution rate and effective population size by region) into paragraphs 3.7 (Substitution rate and spike gene recombination) and 3.8 (Effective population size by region) (revised manuscript lines: 250-261, 265-271).

Round 2

Reviewer 2 Report

Comments and Suggestions for Authors

The manuscript is highly improved. I don't have any further comments.

Comments on the Quality of English Language

The writing is fine.